# The intracellular seven amino acid motif EEGEVFL is required for matriptase vesicle sorting and translocation to the basolateral plasma membrane

Chun-Che Tseng[1], Bailing Jia[1,2], Robert B. Barndt[1], Yang-Hong Dai[3], Yu Hsin Chen[4], Po-Wen A. Du[1,5], Jehng-Kang Wang[5], Hung-Jen Tang[6]*, Chen-Yong Lin[1]*, Michael D. Johnson[1]*

1 Lombardi Comprehensive Cancer Center, Department of Oncology, Georgetown University, Washington, DC, United States of America, 2 Department of Gastroenterology and Hepatology, Henan Provincial People's Hospital, People's Hospital of Zhengzhou University, Zhengzhou, China, 3 Department of Radiation Oncology, Tri-Service General Hospital, Taipei, Taiwan, 4 School of Medicine, National Defense Medical Center, Taipei, Taiwan, 5 National Defense Medical Center, Department of Biochemistry, Taipei, Taiwan, 6 Section of Infectious Diseases, Chi-Mei Medical Center, Tainan, Taiwan

* 8409d1@gmail.com (HJT); lincy@georgetown.edu (CYL); johnsom@georgetown.edu (MDJ)

**Data Availability Statement:** All relevant data are within the manuscript

## Abstract

Matriptase plays important roles in epithelial integrity and function, which depend on its sorting to the basolateral surface of cells, where matriptase zymogen is converted to an active enzyme in order to act on its substrates. After activation, matriptase undergoes HAI-1-mediated inhibition, internalization, transcytosis, and secretion from the apical surface into the lumen. Matriptase is a mosaic protein with several distinct protein domains and motifs, which are a reflection of matriptase's complex cellular itinerary, life cycle, and the tight control of its enzymatic activity. While the molecular determinants for various matriptase regulatory events have been identified, the motif(s) required for translocation of human matriptase to the basolateral plasma membrane is unknown. The motif previously identified in rat matriptase is not conserved between the rodent and the primate. We, here, revisit the question for human matriptase through the use of a fusion protein containing a green fluorescent protein linked to the matriptase N-terminal fragment ending at Gly-149. A conserved seven amino acid motif EEGEVFL, which is similar to the monoleucine C-terminal to an acidic cluster motif involved in the basolateral targeting for some growth factors, has been shown to be required for matriptase translocation to the basolateral plasma membrane of polarized MDCK cells. Furthermore, time-lapse video microscopy showed that the motif appears to be required for entry into the correct transport vesicles, by which matriptase can undergo rapid trafficking and translocate to the plasma membrane. Our study reveals that the EEGEVFL motif is necessary, but may not be sufficient, for matriptase basolateral membrane targeting and serves as the basis for further research on its pathophysiological roles.

**Funding:** This study was supported by National Cancer Institute (NCI) Grant RO1 CA 123223 (to MDJ and CYL), and Grant (MAB-108-079) from the Ministry of National Defense Medical Affairs Bureau, Taiwan and Grants (CMNDMC10705; CMNDMC10813) from Chi-Mei Medical Center, Tainan, Taiwan (to J.-K. Wang). We also acknowledge the assistance provided by the Microscopy and Imaging Shared Resource and the Tissue Culture Shared Resource, which are supported in part by the Lombardi Comprehensive Cancer Center support grant (NIH/NCI grant P30-CA051008). The funders had no role in study design, data collection and analysis, decision to publish, or preparation of the manuscript.

**Competing interests:** CYL is an inventor on US patents #6,077,938 (Title: Monoclonal antibody to an 80-kDa protease) and #6,677,377 (Title: Structure based discovery of inhibitors of matriptase for the cancer diagnosis and therapy by detection and inhibition of matriptase activity) and MDJ and CYL are inventors on US patent #7,355,015 (Title: Matriptase, a serine protease and its applications). This does not alter our adherence to PLOS ONE policies on sharing data and materials.

# Introduction

The type 2 transmembrane serine protease (TTSP) matriptase was initially identified as the major secreted gelatinolytic activity present in conditioned medium from breast cancer cells in an effort to identify and characterize cancer cell-derived extracellular matrix-degrading proteases involved in cancer invasion and metastasis [1,2]. The cloning of the matriptase cDNA and the deduced protein sequence showed that matriptase is a serine protease with trypsin-like activity rather than a matrix metalloprotease (MMPs) [3] but also revealed that matriptase is an integral membrane protein rather than a secreted protein [4,5]. Matriptase targeting to the basolateral plasma membrane was subsequently demonstrated *in vitro* using differentiated polarized epithelial cells and *in vivo* by staining for the protein in human kidney and prostate tissues [6–8]. The purification of activated matriptase in complexes with the Kunitz-type serine protease inhibitor HAI-1 from human breast milk revealed an important functional relationship between matriptase and HAI-1. Furthermore, the targeting of matriptase to the basolateral membrane, combined with its secretion from the apical plasma membrane of lactating mammary epithelial cells, implies a complex and somewhat convoluted cellular itinerary during its life cycle [9].

Like other serine proteases, matriptase is synthesized as a zymogen, which must be converted into an active form by cleavage at the activation motif in order to gain its full enzymatic activity [10]. Under physiological conditions, cell-associated active matriptase is a short-lived species due to its rapid inhibition by HAI-1 through the formation of a stable one-to-one complex [11]. A proportion of the active matriptase is, however, also rapidly shed from the surface of cells [12,13]. HAI-1 is also an integral membrane protein and so can be targeted to the basolateral plasma membrane of polarized epithelial cells [14,15]. Secretion or shedding of matriptase both in the zymogen form and the activated form in complex with HAI-1 from the basolateral plasma membrane has been observed in polarized Caco-2 cells [7], which is conceptually consistent with the expression of matriptase on the basolateral plasma membrane. Interestingly, only activated matriptase in complex with HAI-1 is secreted from the apical plasma membrane of polarized Caco-2, and not the zymogen form of the enzyme [7]. It remains unclear if matriptase can be secreted from the basolateral plasma membrane of cells *in vivo*. The presence of activated matriptase in complex with HAI-1 without measurable matriptase zymogen in human body fluids is consistent with the pattern of matriptase secretion observed *in vitro* in polarized Caco-2 cells [9]. Collectively these *in vivo* and *in vitro* studies illustrate several milestones throughout the matriptase lifespan: 1) synthesis as zymogen, 2) targeting to the basolateral plasma membrane, 3) conversion to an active enzyme and action on its substrates in the basolateral milieu, 4) enzymatic inhibition through the formation of a very stable complex with HAI-1 on the basolateral plasma membrane, 5) internalization of the activated matriptase-HAI-1 complex from the basolateral plasma membrane, 6) transcytosis to the apical face of the cell and 7) shedding from the apical plasma membrane into the lumen of the secretory glands as the activated matriptase-HAI-1 complex, which has been detected in body fluids.

Several molecular mechanisms underlying these milestones in the matriptase lifespan have been well characterized. For example, autoactivation has been identified as the primary mechanism for matriptase zymogen activation [16]. Furthermore, the selective secretion of activated matriptase-HAI-1 complex but not matriptase zymogen from the apical plasma membrane is likely due to the fact that HAI-1 but not matriptase can be internalized from the basolateral surface and undergoes transcytosis to the apical surface [8]. As a consequence, matriptase zymogen on the basolateral surface must be activated and in complex with HAI-1 for secretion from the apical plasma membrane. Thus, much of the regulation and physiological functions

of matriptase must take place on the basolateral plasma membrane, the targeting to which, therefore, represents one of the most important physiological processes in the matriptase life cycle. Basolateral sorting in epithelial cells is mediated by cytoplasmic signals present on membrane proteins. At least three different types of basolateral sorting signal have been identified and characterized, including the tyrosine-based, dileucine, and monoleucine motifs [17]. A cytoplasmic juxtamembrane motif comprised of 6 amino acid residues (45-KQVEKR-50) in rat matriptase was reported to be important for matriptase basolateral sorting [18]. This motif was described by Murai et al., to resemble the sequence responsible for the basolateral sorting found for the rabbit polyimmunoglobulin receptor (pIgR). This sequence does not, however, contain tyrosine or leucine and so probably does not belong to one of the three well-characterized sorting signals. Furthermore, this sequence is not conserved between primate and rodent matriptase: the C-terminal Arg in rat and mouse is replaced by His in the human and chimpanzee proteins. While species variation could explain this difference, the high level of sequence conservation within the matriptase cytoplasmic domain among species suggests that basolateral sorting signals other than the one identified in rat matriptase could be present. In the current study, we revisit the important question as to the nature of the sorting requirement for directing human matriptase to the basolateral plasma membrane.

## Materials and methods

### Cell cultures

HEK293T, the large T expressing variant of the human embryonic kidney line HEK293 (ATCC), HaCaT human keratinocytes (CLS Cell Lines Service GmbH, Eppelheim Germany), and Madin-Darby Canine Kidney (MDCK) cells (ATCC) were cultured in Dulbecco's Modified Eagle Medium (DMEM), supplemented with 10% fetal bovine serum (FBS). The cells were incubated at 37˚C in a humidified atmosphere with 5% $CO_2$.

### Generation of matriptase-EGFP fusion protein

A matriptase fusion protein comprised of 149 amino acids containing the cytoplasmic tail, transmembrane domain, and a partial SEA domain, fused with EGFP at the C terminus was prepared. DNA encoding the partial matriptase MTPN-EGFP sequence flanked by BamH1 and EcoR1 restriction sites were generated by PCR from pCDNA3.1-full-length matriptase [16] using a forward primer: 5′-GGTGAATTCATGGGGAGCGATCGGGC-3′; and a reverse primer: 5′-ACCGGATCCCGCCCTCGCTGAAGGCCGT-3′. The CD-Del-EGFP construct uses a different forward primer: 5′- GGTGAATTCATGGTGGTGCTGGCAGCCGTG-3′ and the same reverse primer as MTPN-EGFP. The PCR product was purified, double digested with EcoR1 and BamHI, and subcloned into the pEGFP-N2 construct (Clontech, Mountain View, CA). The construct was verified by Sanger sequencing and then transfected into HEK293T cells, and expression analyzed 24 hours post transfection by western blot. Other point and deletion mutant constructs were synthesized and subcloned into pEGFP-N1 by General Biosystems (Morrisville, North Carolina).

### Immunoblot

HEK293T cells were lysed in 1% Triton X100 in phosphate-buffered saline containing 1mM 5,5'-Dithio-bis-(2-Nitrobenzoic Acid) (DTNB) to protect the disulfide bonds as previously described [19]. After removal of the insoluble fraction by centrifugation, the protein concentration was determined, and samples containing equal amounts proteins were separated by SDS-PAGE, transferred to nitrocellulose membrane, and probed with matriptase N-terminal

monoclonal antibody PS6. The generation and characterization of the mAb PS6 have been described previously [13].

## Immunofluorescence and live cell microscopy

MDCK cells were plated on coverslips and grown until they became polarized, after which they were transfected with the EGFP fusion construct using Lipofectamine 2000 (Thermo Fischer). The MDCK cells were prepared for staining by washing the coverslips with serum-free DMEM and phosphate-buffered saline (PBS) and then fixed in 10% buffered formalin (Fisher Scientific) for 20 min. The cells on some coverslips were then permeabilized using 0.5% TritonX100 in PBS for 5 min. For studies examining the co-localization of the matriptase-EGFP fusion protein and wild type matriptase, HaCaT cells were incubated with the matriptase mAb M24 at 2 μg/ml at room temperature for 60 min followed by staining with Alexa Fluor 594 goat anti-mouse IgG and for 60 min. ZO-1 was visualized with a ZO-1 antibody (R26.4C from the Developmental Studies Hybridoma Bank, DSHB). R26.4C was deposited to the DSHB by Goodenough, D.A. Phalloidin conjugated to Alexa Fluor 660 was used as a counterstain. Images were captured using a Leica TCS SP8 Laser Scanning Confocal microscope with a 63X oil lens. For live cell imaging, MDCK cells were plated on 30mm glass bottom No.1.0 uncoated culture dishes (MatTek Corp., MA).

## Results

Matriptase is synthesized as a type 2 transmembrane protein, and the transmembrane domain is positioned between a short N-terminal cytoplasmic domain (54 amino acids) and a C-terminal extracellular protein domain known as a SEA domain [4,5]. Matriptase undergoes N-terminal processing by cleavage at Gly-149 within the SEA domain during synthesis and maturation in the ER/Golgi region prior to translocation to the plasma membrane. The matriptase N-terminal fragment is, therefore, comprised of a cytoplasmic domain, transmembrane domain, and half a SEA domain ending at Gly-149. The remaining matriptase extracellular domains, which represent the bulk of the protein and include the serine protease domain, are tethered to the plasma membrane via the non-covalent interactions within the cleaved SEA domain attached to the transmembrane domain. While the basolateral sorting motifs are likely located in the cytoplasmic domain, the transmembrane domain with some portion of the extracellular domains provides the structural basis for insertion of matriptase molecule into the lipid bilayer biomembrane. A green fluorescent reporter protein EGFP was engineered as a proxy for full-length matriptase. EGFP is attached to the carboxyl terminus of the matriptase N-terminal fragment spanning from Met-1 to Gly-149 (Fig 1A). This fusion protein is designated as MTPN-EGFP and can be readily observed as a band at the expected size of approximately 50-kDa detected by the matriptase N-terminal mAb PS6 (Fig 1B).

## The cytoplasmic domain is required for matriptase targeting to the plasma membrane

When the MTPN-EGFP was expressed in the HaCaT human keratinocytes, the fusion protein was observed at the cellular periphery, primarily at the contact areas between neighboring cells (Fig 2A). A significant MTPN-EGFP signal was also observed as punctate staining throughout the cells, particularly focused in the perinuclear area. This punctate staining profile likely results from the proportion of the fusion protein in the intracellular synthetic and trafficking pools. Endogenous matriptase, as demonstrated by indirect immunofluorescence staining, resembled MTPN-EGFP with regard to its cellular distribution profile with signal observed both on the cell periphery and in the intracellular pools (Fig 2B). When these images were

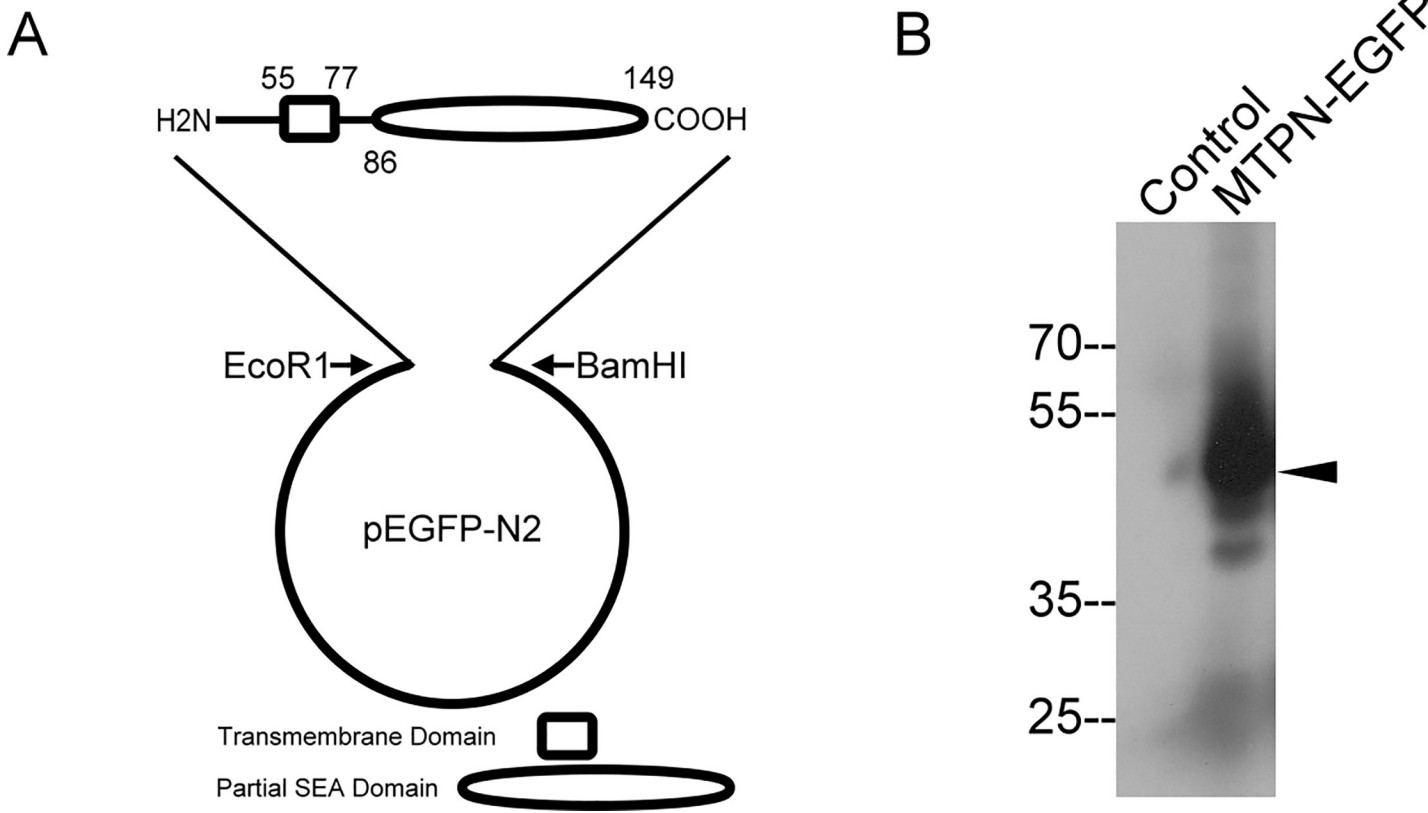

**Fig 1. Schematic diagram and expression of the matriptase N-terminal fragment-green fluorescent protein (EGFP) fusion protein.** A. The matriptase N-terminal fragment containing the short cytoplasmic domain, transmembrane domain (aa 55–77), and half of the SEA domain (aa 86–149) was inserted into pEGFP expression vector via EcoR1 and BamH1 sites. B. This construct, named MTPN-EGFP, was transiently expressed in HEK293T cells. Lysates prepared from the HEK293T parental cells as a negative control (Control) and from MTPN-EGFP-expressing (MTPN-EGFP) HEK293T cells were analyzed by immunoblot analysis using the matriptase mAb PS6 to identify the matriptase N-terminal fragment-EGFP fusion protein. The fusion protein was identified as a band at the predicted molecular mass of approximately 50-kDa.

merged with staining for F-actin and DNA (nuclei) (Fig 2C), significant overlap between the endogenous matriptase staining and GFP signal from the fusion construct was observed. These data suggest that MTPN-EGFP behaves like endogenous matriptase concerning its translocation to the plasma membrane and accumulation at cell-cell junctions (Fig 2A, 2B and 2C, white arrows). While some intracellular MTPN-EGFP may also be co-localized with endogenous matriptase (Fig 2A, 2B and 2C, pink arrows), MTPN-EGFP was seen in granule/vesicle-like structures much larger than those observed for endogenous matriptase. These different intracellular localization patterns could mean that MTPN-EGFP traffics to the plasma membrane via different routes than endogenous matriptase. However, it may be that the exogenous fusion protein tends to accumulate in these compartments more than endogenous matriptase resulting in some secretory vesicles being very bright for MTPN-EGFP. Nevertheless, the matriptase N-terminal fragment, including the intracellular domain, transmembrane domain, and the N-terminal portion of the cleaved SEA domain, contains the structural requirements for cell surface and the cell-cell junction translocation. It is worth noting that the matriptase mAb M24 used for the detection of endogenous matriptase recognizes an epitope on the fourth LDL receptor class A domain, which is not part of the MTPN-EGFP construct.

The importance of the cytoplasmic domain in membrane targeting, which is a prerequisite for basolateral sorting, was next examined by generating a construct in which the cytoplasmic

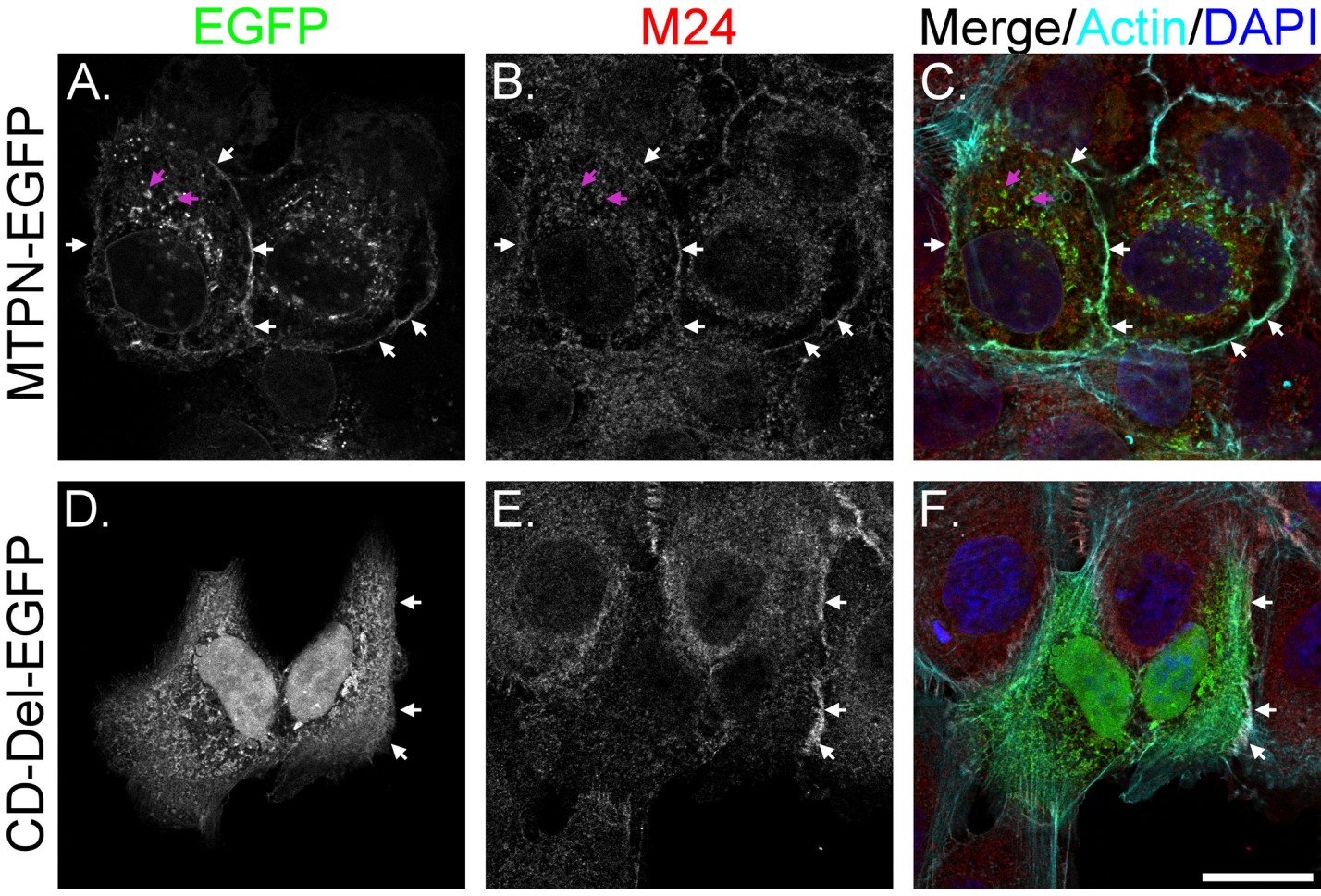

**Fig 2. Matriptase cytoplasmic domain is required for targeting to the cell periphery.** Both the wild-type (MTPN-EGFP) and cytoplasmic domain deleted (CD-Del-EGFP) matriptase-EGFP constructs were transiently expressed in HaCaT human keratinocytes. The subcellular localization of both MTPN-EGFP (A and C) and CD-Del-EGFP (D and F) were compared with the endogenous matriptase, which was analyzed by indirect immunofluorescent staining with the matriptase-specific mAb M24 (B and E) followed by Alexa 594-labeled anti-mouse IgG. The cells were also stained for F-actin using Alexa 660-labeled phalloidin (C and F, Cyan) and nuclei using DAPI (C and F, blue), as counterstains. The staining is presented as black and white images (A, B, D, and E) and as merged false-color images (C and F). The white arrowheads indicate cell periphery and red arrowheads indicate intracellular vesicles. Scale bar: 10 μm.

domain (amino acids Gly2-Arg54) from the MTPN-EGFP had been removed (CD-Del-EGFP). It is worth noting that there is no canonical signal motif in matriptase and the N-terminal hydrophobic stretch is considered to function as the signal peptide and transmembrane domain. When transiently expressed in HaCaT cells, the CD-Del-EGFP was observed to accumulate in the nucleus and perinuclear area (Fig 2D). In cells expressing the CD-Del-EGFP, endogenous matriptase staining was unchanged with signal at the cell periphery (Fig 2E, arrows) and also scattered throughout the cells as a fine punctate staining pattern, similar to that in other neighboring cells not expressing CD-Del-EGFP. The lack of localization of the CD-Del-EGFP construct at the cell periphery (Fig 2D, 2E and 2F) suggests that the cytoplasmic domain is essential for plasma membrane targeting in spite of the presence of the transmembrane domain. The altered intracellular distribution profile of the CD-Del-EGFP construct, particularly the apparent nuclei localization after the removal of the cytoplasmic domain, suggests that the cytoplasmic domain is important for the sorting and entry of matriptase into the correct secretory vesicles en route to the plasma membrane. In short, we generated

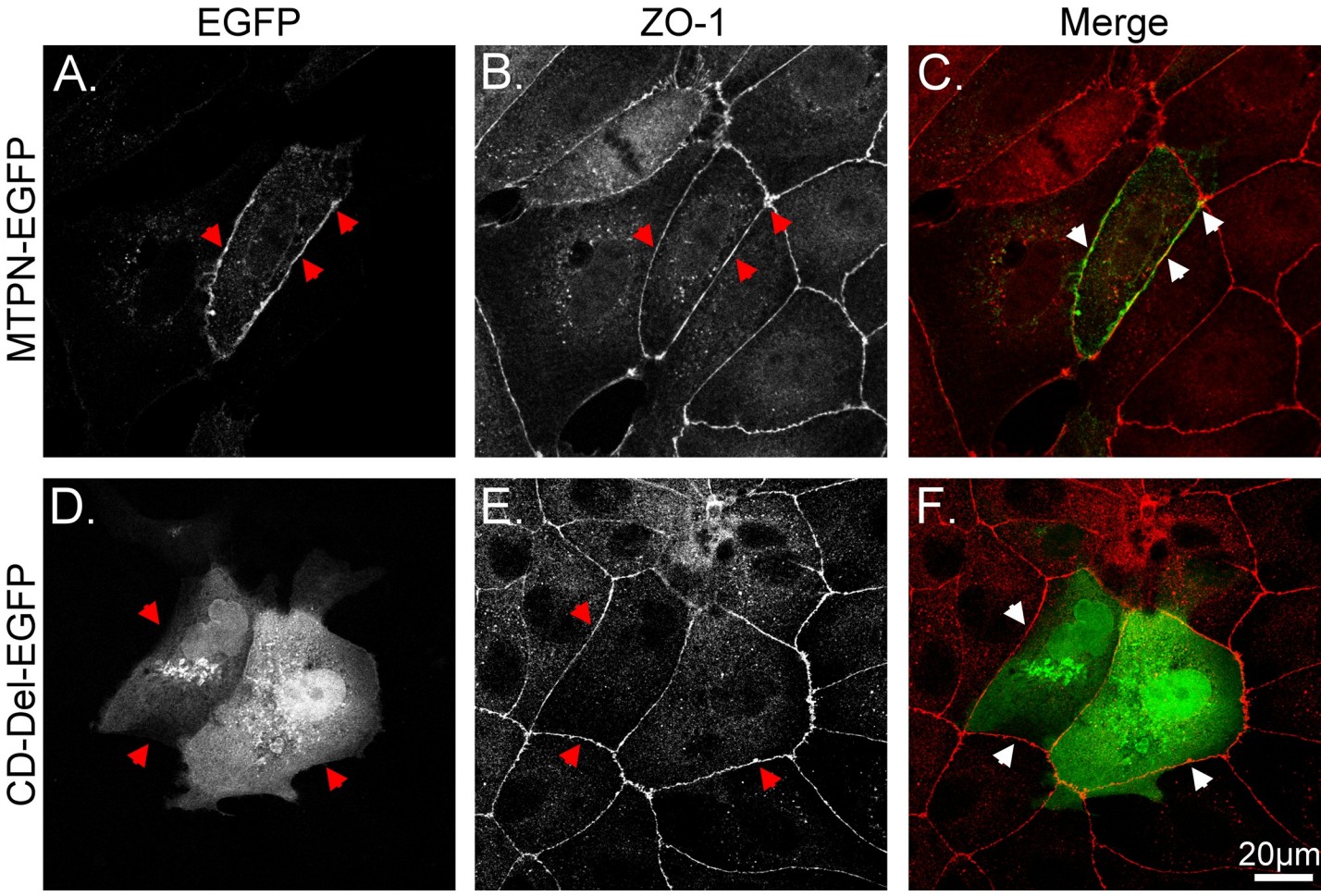

**Fig 3. Exogenous expression and subcellular localization of MTPN-EGFP and CD-Del-EGFP in MDCK cells.** Both wild-type (MTPN-EGFP) and cytoplasmic domain deleted (CD-Del-EGFP) matriptase-EGFP constructs were transiently expressed in MDCK canine distal renal tubular epithelial cells. The subcellular localization of both MTPN-EGFP (A and C) and CD-Del-EGFP (D and F) were compared with the tight junction marker ZO-1, which was analyzed by indirect immunofluorescent staining using ZO-1 antibody, followed by Alexa 594-labeled anti-mouse IgG (B and E). The staining is presented as black and white images (A, B, D, and E) and merged false-color images (*C* and *F*). The arrowheads indicate the cell periphery. Scale bar: 20 μm.

MTPN-EGFP, which resembles endogenous matriptase with respect to trafficking and translocation behavior, and demonstrated that the matriptase cytoplasmic domain is vital to appropriate trafficking and membrane targeting.

MDCK cells have been widely used to model the polarization of simple epithelial cells. When grown at confluency, these cells form a polarized monolayer with well-defined tight junctions and provide an ideal system to investigate the molecular determinants for polarized plasma membrane targeting and trafficking. We, therefore, next used these canine distal renal tubular epithelial cells as a polarized model to study the plasma membrane translocation and to help us identify the matriptase basolateral sorting determinants using MTPN-EGFP and its derivatives (Figs 3–6). When transiently expressed in MDCK cells, MTPN-EGFP was observed both on the cell periphery and inside the cells (Fig 3A) with the distribution profile almost exactly the same as that in HaCaT cells (Fig 2). The peripheral staining was further confirmed by its coincidence with ZO-1, a well-established tight junction marker (Fig 3A, 3B and 3C, as indicated by arrowheads). The loss of plasma membrane localization, the aberrant intracellular distribution, and nuclear localization observed in HaCaT cells for the CD-Del-EGFP construct

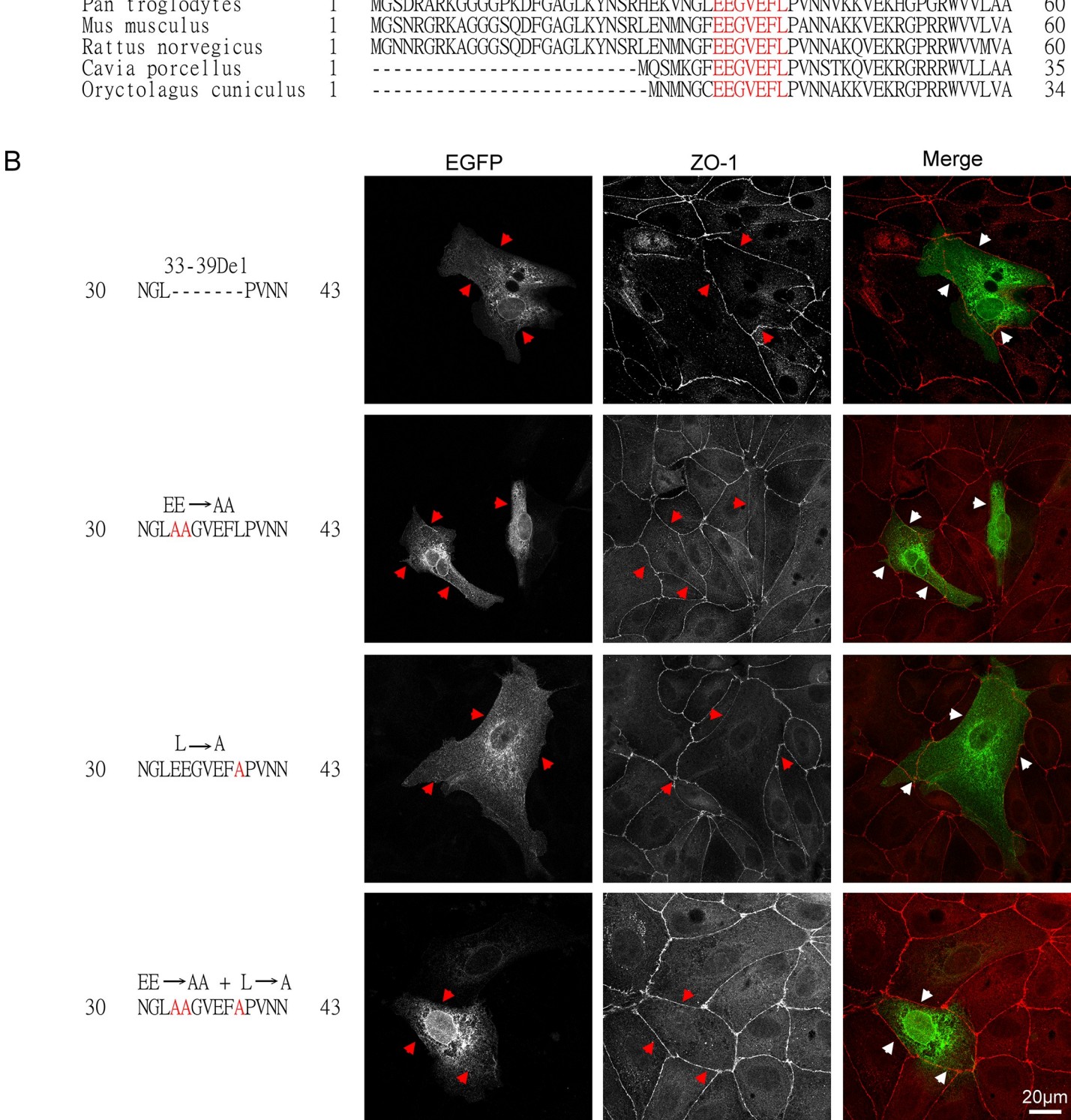

**Fig 4. The seven amino acid residues between residues 33 and 39 are required for matriptase targeting to the cell periphery.** A. The amino acid sequence of the matriptase cytoplasmic domain between residues 1 and 50 were compared among different species, including human (Homo sapiens), common chimpanzee (Pan troglodytes), house mouse (Mus musculus), common rat (Rattus norvegicus), guinea pig (Cavia porcellus), and rabbit (Oryctolagus cuniculus). The seven amino acid residues EEGVEFL (33–39 in human matriptase) are highlighted in red. B. The four mutated matriptase-EGFP constructs, including 33-39Del, EE→AA, L→A, and EE→AA + L→A, as indicated, were transiently expressed in MDCK cells. The subcellular localization of these matriptase-EGFP fusion proteins was compared with that

of the tight junction marker ZO-1, which was analyzed by indirect immunofluorescent staining using a ZO-1 antibody, followed by Alexa 594-labelled anti-mouse IgG. The staining is presented as black and white images (EGFP and ZO-1) and merged false-color images (Merge). The arrowheads indicate the cell periphery. Scale bar: 20 μm.

were also observed in MDCK cells (Fig 3D, 3E and 3F). Collectively, these data confirm the role of matriptase cytoplasmic domain in the plasma membrane targeting and validate the MDCK system as a model to study matriptase basolateral targeting with the matriptase N-terminal fragment-EGFP fusion protein.

## Monoleucine C-terminal to an acidic cluster is required for matriptase basolateral targeting

By comparing the sequence of the matriptase cytoplasmic domain with the known basolateral sorting signals, we noted that the six amino acid residues 33-EEGEVFL-39 are very similar to the basolateral sorting motif of the type monoleucine C-terminal to an acidic cluster, EEXXXL

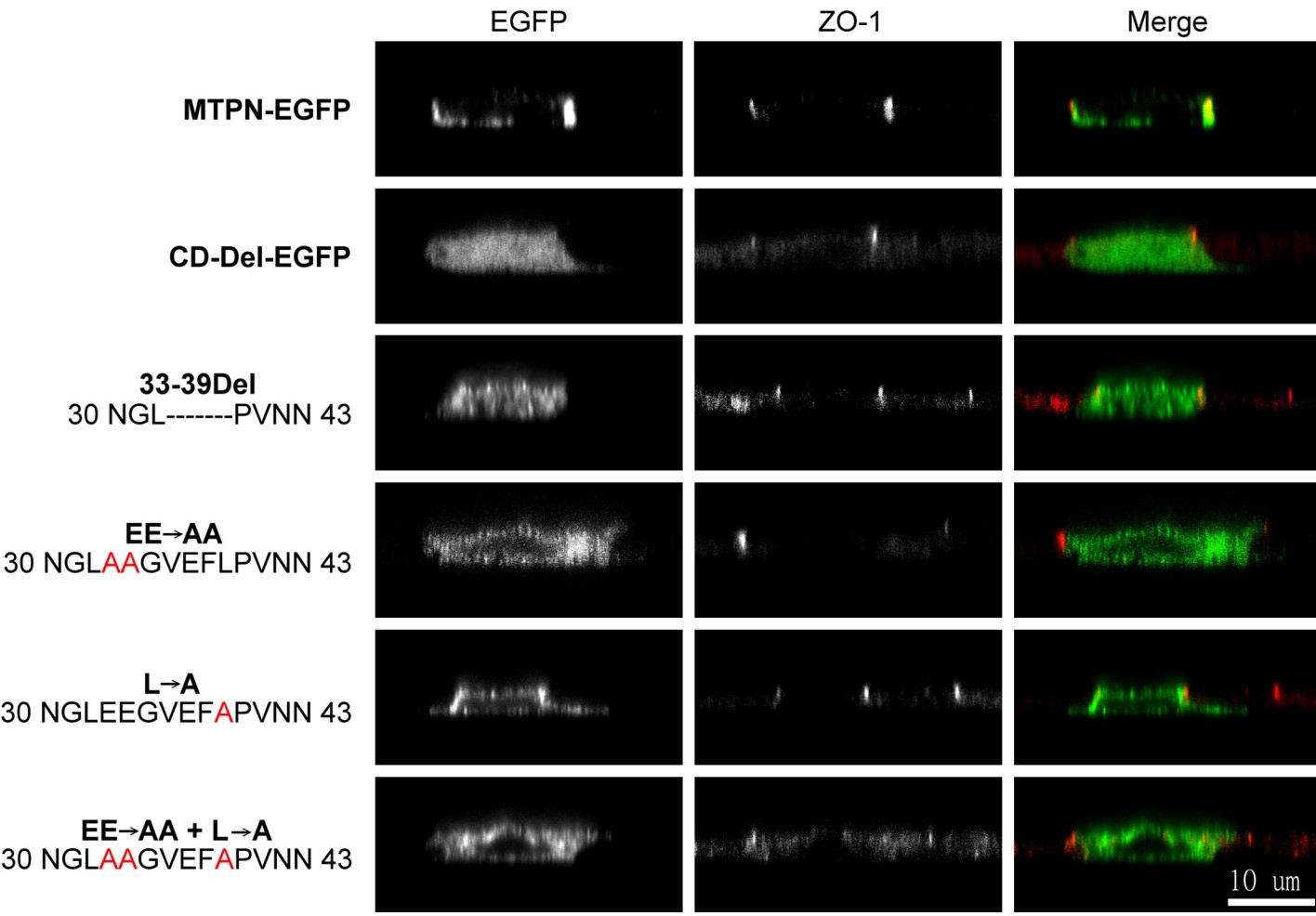

**Fig 5. The motif consisting of residues 33–39 with monoleucine C-terminal to acidic cluster contributes to matriptase basolateral sorting.** The matriptase-EGFP constructs, including MTPN-EGFP, CD-Del-EGFP, 33-39Del, EE→AA, L→A, and EE→AA + L→A, as indicated, were transiently expressed in MDCK cells. The subcellular localization and X-Z distribution of these matriptase-EGFP fusion proteins were compared with that of the tight junction marker ZO-1, which was analyzed by indirect immunofluorescent staining using ZO-1 antibody, followed by Alexa 594-labeled anti-mouse IgG. The staining is presented as black and white images (EGFP and ZO-1) and merged false-color images (Merge). Scale bar: 10 μm.

## MTPN-EGFP  CD-Del-EGFP  EE→ AA + L→A  33-39Del

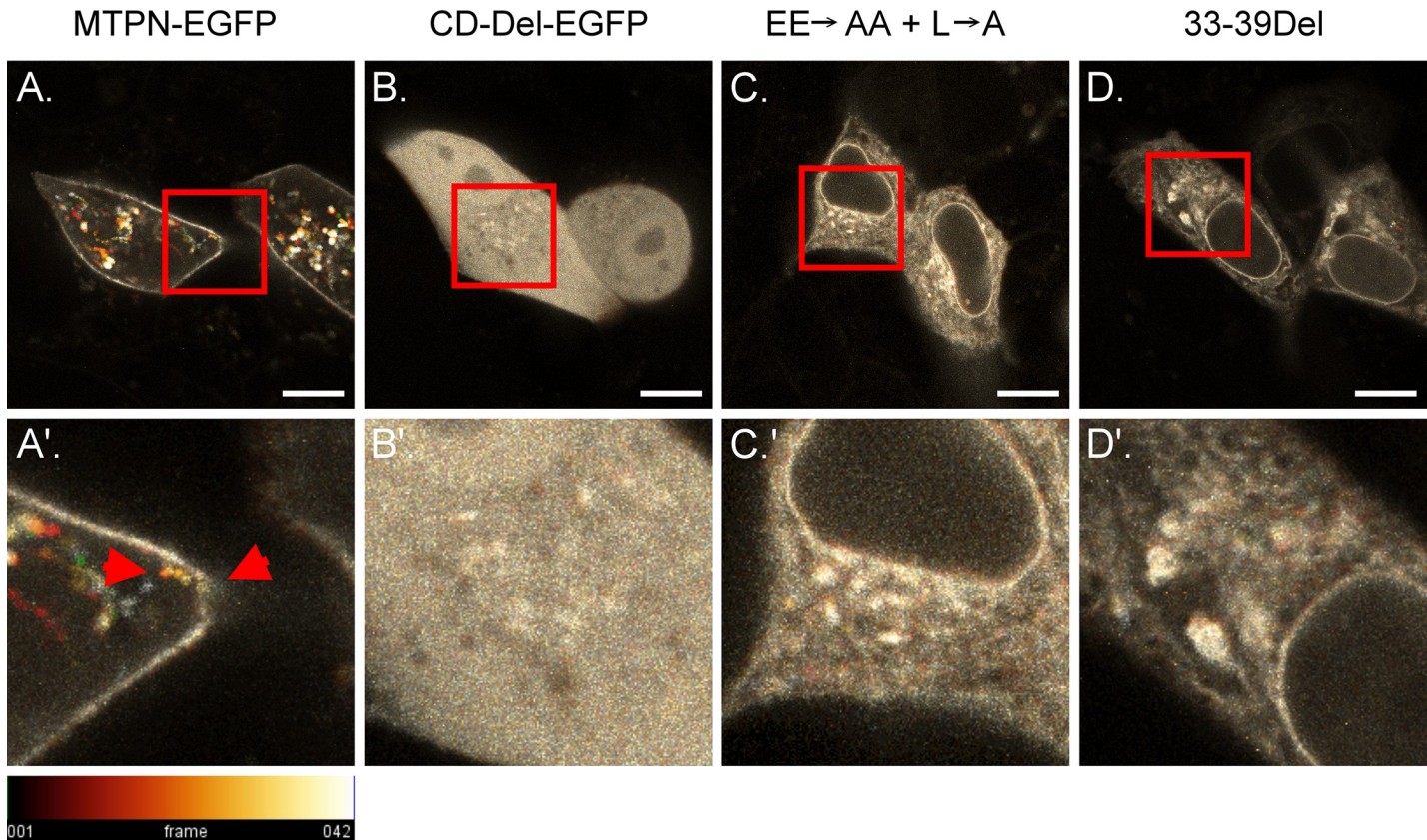

**Fig 6. Monoleucine sorting motif mutants lose subcellular polarity and are retained in the ER/Golgi secretory pathway.** Polarized MDCK cells transiently expressing the MTPN-EGFP, CD-Del-EGFP, and the monoleucine sorting motif mutants, the EE→AA+L→A and 33-39Del, as indicated, were cultured on glass-bottom dishes and imaged live at 37C° 5% $CO_2$ by laser scanning confocal microscopy. Forty (40) seconds of footage are presented using temporal-color indexing. A vesicle containing MTPN-EGFP was observed fusing with the membrane as indicated by the trail of color transitioning from orange to white, as indicated by the two red arrowheads. Bar, 10 μm.

[20–22]. Furthermore, this seven amino acid matriptase motif is highly conserved across different species (Fig 4A, in red). We, therefore, set out to determine whether the EEGEVFL sequence might be required for matriptase basolateral sorting by examining the impact of alterations in the sequence on the subcellular and basolateral distribution (Figs 4 and 5). The seven amino acids were either deleted entirely from the MTPN-EGFP construct (33-39Del; NGL——-PVNN), or point mutations were made of the characteristic C-terminal Leu and the two N-terminal Glu residues either alone or combination (L→A; EE→AA; EE→AA + L→A). These four constructs were then expressed in polarized MDCK cells to see whether and how the modifications might perturb matriptase distribution and subcellular localization.

The deletion of all seven amino acid residues apparently prevented the fusion protein from translocation to the cell periphery as the EGFP was concentrated in the middle of the cells in a finely punctate pattern (Fig 4B, 33-39Del). Similar GFP patterns were also observed for the EE→AA, the L→A, and the EE→AA + L→A mutants (Fig 4B). The lack of significant accumulation of these mutants at the cell periphery was further validated by an attempt to assess the co-localization of these mutants with the tight junction marker ZO-1 (Fig 4B, indicated by the arrowheads), while the presence of inconspicuous yet discernible green fluorescent signal patches was occasionally seen in a small portion of the cells. Thus, while these mutants appear to be synthesized and traffic through the secretory pathway due to their perinuclear

accumulation, the lack of accumulation at the cell periphery suggests that these three characteristic amino acid residues play important roles in the sorting and trafficking of matriptase.

The XZ distribution of the MTPN-EGFP constructs and its mutants in polarized MDCK cells was next examined (Fig 5). The targeting of MTPN-EGFP to cell periphery, including the accumulation at the contacts between neighboring cells, on the XY axis (Fig 3) was further narrowed to the basal and lateral plasma membrane as the EGFP signal was seen primarily on the cell-substratum and cell-cell contacts beneath the apical marker ZO-1 (Fig 5, MTPN-EGFP). In contrast, the EGFP signal from the construct in which the cytoplasmic domain is deleted (CD-Del-EGFP) was homogeneously distributed throughout the cells and not present in the three subdomains of the plasma membrane (Fig 5, CD-Del-EGFP). These data confirm the role of the cytoplasmic domain in matriptase basolateral targeting. The deletion of the seven amino acid motif also prevented the EGFP signal from being targeted to the plasma membrane (Fig 5, 33-39DEL), further narrowing down the role of these the seven amino acids in the cell surface targeting and likely the basolateral plasma membrane targeting. It is worth noting that the deletion of the seven amino acid motif did not appear to affect the entrance of the 33-39Del construct into secretory vesicles as the EGFP signal was present in vesicles rather than homogenously distributed throughout the cells seen for the CD-Del-EGFP. The importance of the C-terminal leucine and the double glutamic acid cluster within the 7 amino acid motif was demonstrated by the significant EGFP signal observed on entire cell periphery, including the apical plasma membrane, for the constructs bearing these mutations alone or in combination (Fig 5, EE→AA, L→A, and EE→AA + L→A). In contrast to the deletion of the entire motif, mutation of these three amino acid residues did not appear to affect translocation to the cell periphery. Collectively, these data support the role of the seven amino acids motif with the characteristic leucine and glutamic acid cluster in matriptase basolateral sorting.

Basolateral targeting signals facilitate protein delivery by virtue of their interaction with sorting associated proteins, such as the Rab small GTPases. These interactions are important for the trafficking of vesicles targeting to specific subcellular locations. The role of the seven amino acid motif in matriptase basolateral sorting was further investigated by monitoring EGFP-containing trafficking vesicles using time-lapse live-cell imaging (S1 Video to S4 Video) and temporal-color codes in hyperstack projections (Fig 6). Hyperstack projection of temporal-color coding was used to project trafficking vesicles at various time points onto a single image, generated using the Image J script hosted at GitHub (Fiji, https://github.com/fiji/fiji/blob/master/plugins/Scripts/Image/Hyperstacks/Temporal-Color_Code.ijm). In temporal-color coding the location of a moving vesicle is indicated by different colors marking the track of the vesicle, whereas those that are not moving are represented only by the color of the last time point. MTPN-EGFP can be seen on the cell periphery and in distinct trafficking vesicles, which were moving dynamically, appearing, disappearing, and fusing with the plasma membrane (S1 Video). When viewed in the Hyperstacks, the plasma membrane-associated MTPN-EGFP largely remained still and was seen in the cell periphery in white (Fig 6A). The trafficking vesicles containing MTPN-EGFP were seen in different colors due to their dynamic movement. The fusion of an MTPN-EGFP-containing vesicle to the plasma membrane is illustrated by tracking a vesicle with rapidly changing color from orange to yellow and then to white as it approaches the cell periphery (Fig 6A', MTPN-EGFP, between the two arrowheads). In contrast to the plasma membrane localization and the dynamic movement of distinct vesicles containing MTPN-EGFP, locations marked by the CD-Del-EGFP construct appeared stationary throughout the cells (S2 Video). Viewed as Hyperstacks, the lack of movement renders the CD-Del-EGFP white (Fig 6B and 6B'). The deletion of the entire motif (33-39Del) or the combined mutations of the characteristic Leu and glutamic acid cluster (EE→AA + L→A) apparently compromised the rapid movement of the trafficking vesicles (S3 and S4 Videos).

These mutations also prevented the EGFP signal from accumulating at the cell periphery. Instead, the EGFP signal from the two mutants accumulated on the periphery of the nuclei. Viewed as Hyperstacks, the limited movement rendered both mutants in white (Fig 6C, 6C', 6D and 6D'). This reduced movement demonstrates the importance of the motif and the three characteristic amino acid residues in matriptase sorting and suggests it likely lies in their roles in vesicle entry and trafficking.

## Discussion

Analysis of the subcellular localization and trafficking of the matriptase N-terminal fragment fused with EGFP (and mutants of the construct) in HaCaT and polarized MDCK cells, allowed us to identify a protein motif within the cytoplasmic domain that appears to be required for matriptase basolateral targeting. The matriptase motif consists of 7 amino acid residues (33-EEGVEFL-39) and resembles the signal motifs recently discovered in amphiregulin (236-EERKKL-241) [20] and heparin binding EGF (197-EEKVKL-201) [22]. A similar but as yet unconfirmed basolateral sorting motif is also present in betacellulin (157-EEMETL-163) [21]. These sorting motifs feature a monoleucine C-terminus linked to an acidic cluster. There is one acidic residue in the motifs present in matriptase and betacellulin, compared to 3 and 2 basic residues present in amphiregulin and HB-EGF, respectively. Since the two lysine residues in the amphiregulin motif are not required for basolateral sorting, the charge and/or size of the amino acid residues outside of the characteristic leucine residue and the acidic cluster may not be important for basolateral sorting. Likewise, the four amino residues found between the acidic cluster and the leucine in the matriptase motif *versus* three residues in the three EGF ligands indicate some flexibility in the length of this region of the sorting motif.

The requirement for the 33-EEGVEFL-39 motif in basolateral sorting identified in the current study adds a new molecular determinant to the set of known physiological mechanisms governing the function and enzymatic activity of matriptase, which is expressed as a mosaic protein with several characteristic structural domains and has a life cycle that involves a complex subcellular journey. Targeting to the basolateral plasma membrane through this newly identified intracellular motif represents an early hallmark in matriptase life cycle. At the basolateral face of polarized epithelial cells, matriptase proteolytic activity can act on substrates of either epithelial or stroma origins. These putative substrates include stroma-derived hepatocyte growth factor (HGF) as well as protease activated receptor (PAR)-2 which is of epithelial cell origin [23,24]. Prior to translocation to the basolateral plasma membrane, matriptase undergoes an unusual post-translational modification, with N-terminal processing mediated by the autolytic cleavage of the SEA domain [13,25]. The key amino acid involved in this event has been identified as Gly-149. N-terminal processing is important and is required for matriptase zymogen activation at a later stage of its life cycle [16], but may not have a significant role in the basolateral sorting of the protein. This is implied by the fact that the MTPN-EGFP construct used in the current study contains the matriptase N-terminal fragment from Met-1 to Gly-149 and so only contains half of the SEA domain. Other important molecular determinants have been also identified and shown to be involved in the control of zymogen activation and/or directly linked to matriptase enzymatic activity. For example, His-656, Asp-711, and Ser-805 in the active site triad and Asp-799 in the substrate binding pocket are responsible for the potent trypsin-like activity of matriptase [3–5]. In additional to their role in the enzymatic activity of the mature enzyme, the three residues in the active site triad are also important for matriptase intrinsic zymogen activity, which is responsible for mediating matriptase autoactivation [16]. Cleavage of Arg-614 within the activation motif, and the Asp residues in the calcium cages of the four LDL receptor class A domains (D482, D519, D555, and D598) are also

important for matriptase zymogen activation. Cleavage at Arg-186, which is temporally linked with matriptase zymogen activation and results in matriptase shedding from the plasma membrane, appears to be the starting point for the removal of matriptase from the basolateral surface. After this cleavage, the enzyme is either shed into the extracellular milieu from the basolateral surface, or is inhibited by HAI-1 through the formation of a very stable protease-inhibitor complex. The HAI-1-matriptase complex is subsequently internalized and undergoes transcytosis for secretion from the apical cell surface [13]. Understanding the role of these structural domains and these critical residues provides insight into the molecular mechanisms by which matriptase is regulated.

The peri-nuclear localization of CD-Del-EGFP construct and the EE→AA + L→A mutants might raise technical and mechanistic concerns regarding the cause of this localization. Although one could attribute the aberrant subcellular localization to potentially artefactual effects, the deletion and point mutations that abolish matriptase translocation to the plasma membrane or prevent the majority of the point mutated MTPN-GFP variant from translocating to the basolateral plasma membrane. This result supports the important role of the cytoplasmic tail in matriptase plasma membrane targeting and these seven amino acid residues in the basolateral targeting, regardless of the mechanism underlying the loss of plasma or basolateral membrane trafficking. Some GPI-anchored proteins, such as testisin and prostasin, contain no or a very short cytoplasmic tail and can undergo co- or post-translational modifications and subsequent translocation to the plasma membrane. Furthermore, these EGFP-containing proteins emit green fluorescent signal, and thus they do undergo protein synthesis and folding in at least a somewhat "normal" manner. The loss of plasma membrane translocation may, thus, result from preventing CD-Del-EGFP from entering either the endoplasmic reticulum (ER) or the secretory pathway. The former would mean that CD-Del-EGFP is synthesized in the cytosol rather than the ER, whereas the latter would suggest the important role of the intracellular tail in the cell surface translocation. The misrouted subcellular localization of CD-Del-EGFP has, nevertheless, its molecular and cellular basis. Similarly, the entrance into nuclei or the accumulation at the nuclear periphery for the EE→AA + L→A, the 33–39 DEL, and the EE→AA mutants, also cannot simply be interpreted as artefactual effects. The lack of the accumulation at the nuclear periphery and the nuclear localization for the L→A mutant supports the technical feasibility and soundness of the deletion and point mutations on the seven amino acid motif. The altered subcellular distribution and intracellular trafficking associated with the deletion and point mutations of the seven amino acid motif, therefore, demonstrate the important role of the motif in matriptase vesicle sorting and translocation to the basolateral plasma membrane.

In summary, a protein motif, which is conserved among different species and which contains seven amino acid residues (33-EEGVEFL-39) located within the cytoplasmic domain, was identified and shown to be required for matriptase basolateral targeting. This motif is primarily involved in the movement of matriptase-containing vesicles. The matriptase sorting motif resembles those motifs with a monoleucine C-terminus and an acidic cluster, which was initially identified in some members of the EGF ligand family. Our current study not only identifies an important molecular determinant governing an important physiological function of matriptase but also expands the role of the class of motif with a monoleucine C-terminus to an acidic cluster in basolateral targeting from EGF family ligands to a type 2 transmembrane serine protease.

## Supporting information

**S1 Video. (MTPN-EGFP): Movie time stamp beginning at 02:51.570 (frame 134), ending at 03:44.460 (frame 175).**
(WMV)

**S2 Video. (CD-Del-EGFP): Movie time stamp beginning at 00:00.000 (frame 1), ending at 00:52.890 (frame 42).**
(WMV)

**S3 Video. (EE→AA+L→A): Movie time stamp beginning at 01:05.790 (frame 52), ending at 01:58.680 (frame 93).**
(WMV)

**S4 Video. (33-39Del): Movie time stamp beginning at 00:00.000 (frame 1), ending at 00:52.890 (frame 42).**
(WMV)

## Author Contributions

**Conceptualization:** Chun-Che Tseng, Jehng-Kang Wang, Chen-Yong Lin, Michael D. Johnson.

**Data curation:** Chun-Che Tseng, Bailing Jia.

**Formal analysis:** Chun-Che Tseng, Bailing Jia, Yang-Hong Dai, Yu Hsin Chen, Po-Wen A. Du.

**Funding acquisition:** Jehng-Kang Wang, Hung-Jen Tang, Michael D. Johnson.

**Investigation:** Bailing Jia, Robert B. Barndt, Yu Hsin Chen, Po-Wen A. Du.

**Methodology:** Chun-Che Tseng, Robert B. Barndt.

**Supervision:** Michael D. Johnson.

**Validation:** Chun-Che Tseng.

**Writing – original draft:** Chun-Che Tseng, Robert B. Barndt, Jehng-Kang Wang, Chen-Yong Lin, Michael D. Johnson.

**Writing – review & editing:** Chen-Yong Lin, Michael D. Johnson.

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
