## [Decision Letter · Decision Letter 0]

17 Jul 2019

PONE-D-19-15523

Human matriptase is sorted to the basolateral plasma membrane via a motif with monoleucine C-terminal to an acidic cluster

PLOS ONE

Dear Dr. Chen-Yong Lin,

Thank you for submitting your manuscript to PLOS ONE. After careful consideration, we feel that it has merit but does not fully meet PLOS ONE’s publication criteria as it currently stands. Therefore, we invite you to submit a revised version of the manuscript that addresses the points raised during the review process.

We would appreciate receiving your revised manuscript by August 20, 2019. To enhance the reproducibility of your results, we recommend that if applicable you deposit your laboratory protocols in protocols.io, where a protocol can be assigned its own identifier (DOI) such that it can be cited independently in the future. For instructions see: http://journals.plos.org/plosone/s/submission-guidelines#loc-laboratory-protocols

We look forward to receiving your revised manuscript.

Kind regards,

Qing-Xiang Amy Sang, Ph.D.

Academic Editor

PLOS ONE

Journal Requirements:

3. We note that you have a patent relating to material pertinent to this article. Please provide an amended statement of Competing Interests to declare this patent (with details including name and number), along with any other relevant declarations relating to employment, consultancy, patents, products in development or modified products etc. Please confirm that this does not alter your adherence to all PLOS ONE policies on sharing data and materials, as detailed online in our guide for authors http://journals.plos.org/plosone/s/competing-interests by including the following statement: "This does not alter our adherence to  PLOS ONE policies on sharing data and materials.” If there are restrictions on sharing of data and/or materials, please state these. Please note that we cannot proceed with consideration of your article until this information has been declared.

Reviewers' comments:

Reviewer's Responses to Questions

**Comments to the Author**

1. Is the manuscript technically sound, and do the data support the conclusions?

Reviewer #1: Partly

Reviewer #2: Yes

2. Has the statistical analysis been performed appropriately and rigorously? 

Reviewer #1: Yes

Reviewer #2: Yes

3. Have the authors made all data underlying the findings in their manuscript fully available?

Reviewer #1: Yes

Reviewer #2: Yes

4. Is the manuscript presented in an intelligible fashion and written in standard English?

Reviewer #1: No

Reviewer #2: Yes

5. Review Comments to the Author

Reviewer #1: The aim of the study was to identify and characterize a cytoplasmic basolateral sorting signal in the transmembrane serine protease, matriptase. In fact, a basolateral sorting motif in the protein, which resembles the sequence responsible for basolateral sorting of the polymeric immunoglobulin receptor, has been identified in previous studies. However, the authors have decided to revisit the question since that signal does not contain canonical tyrosines or leucines, and is not conserved among species. Since the enzyme is an important one, and since polarized sorting could play an important role in its function in health and disease, the research aim is important and relevant.

Using site-directed mutagenesis and cell biological (mainly morphological)-based assays, the authors propose that a basolateral sorting motif that is based on leucine and an acidic cluster exists in the cytoplasmic domain of matriptase. Notably, all research was performed on a matriptase N-terminal fragment ending at amino acid Gly-149 fused at its extracellular C-terminus to eGFP.

Critiques:

I feel that the reported data do not firmly imply that the identified signal is indeed a basolateral sorting signal. Therefore, a significant amount of research has to be invested to address this point properly.

1). The importance of Figs 1 & 2 is to show that the ectopically expressed fusion protein localization is approximately normal. This was done by comparing its localization with respect to the endogenous protein (stained with the M24 antibody that recognizes the endogenous, but not the ectopically expressed protein) in Fig. 1, and by showing the preferential plasma membrane localization of the expressed protein with respect to ZO-1 staining, in Fig. 2. I think that the data in Fig.2 are not important for the studied question. In addition, deletion of cytoplasmic tails of transmembrane proteins typically have gross and often times artifactual effects on membrane trafficking of the mutated protein.

2). Regarding data presented in Fig. 4. The resolution of this image (as well as of other images) was typically low, posing a difficulty to understand the details that the authors have pointed out in the text. For instance, I did not get their interpretation that the regions marked with arrows and appear in green mark the cell periphery. The different mutants presented show ER, or possibly some Golgi-like staining. In fact, the subtitle given to the related Figure 6 may support this idea. But, since there was not an attempt to address this point, it is hard to say where these mutants are located within the cells. I would also suggest adding the MTPN-wt distribution to the composite to facilitate the comparison. Nonetheless, this is an important point because if indeed the protein is stuck in the early secretory pathway, then the basolateral sorting signal is not at all such a signal, but an ER or Golgi localization signal. Typically, inactivation of a basolateral sorting signal causes the mislocalization of a protein to the apical surface and has no significant impact on the exit of the mutated protein from the ER or Golgi.

3). As per the experiments described in Fig. 5, which assessed the effects of the mutated residues on the apical and basal localization of the protein in polarized MDCK cells. To verify that the mutated versions of the protein are indeed mislocated to the apical surface, the authors have to perform genuine surface stainings of the protein, using at least one of the two indicated approaches; a) performance of surface labeling of the protein using antibodies directed against the ectodomain of the protein, and under conditions whereby the cells are not permeabilized. In this strategy, providing confocal x-z images is not sufficient, and quantitative analysis of the Ap vs. Bl distribution of the protein should be provided. b) performance of cell-surface biotinylation/biochemical based assay in which cell surface proteins are biotinylated and pulled down and the protein of interest is detected by Western blotting. These experiments are fundamental for the proper addressing the effects of the mutations on the polarized distribution of matriptase.

4). A prominent hallmark of basolateral sorting signals is that they can act in a dominant and autonomous manner. Thus, it would be essential to show that the signal identified in this study can confer basolateral sorting of a heterologous protein that is intrinsically sorted to the apical surface, and that the mutations abrogate this process.

5). I found the paper difficult to read, containing unexplained jargon and abbreviations, s as well as long and unclear sentences (e.g., in page 11 the sentence starting with “ The lack of significant…”, or “The targeting of MTPN-EGFP to cell periphery…”).

Reviewer #2: Human matriptase is sorted to the basolateral plasma membrane via a motif with

monoleucine C-terminal to an acidic cluster

In this manuscript, Tseng CC et al addressed the question of the basolateral sorting motif for human matriptase. They identified a monoleucine C-terminal to an acidic cluster in the amino terminus of matriptase to determine matriptase sorting to the basolateral plasma membrane. The authors used a fusion protein containing a green fluorescent protein (EGFP) reporter to conjugate with the ending residue G149 in the amino terminus of matriptase. By using this fusion reporter protein, they reveal the basolateral sorting motif for matriptase. However, several issues are needed to be clarified.

1. The red and green fluorescence are not well shown in the individual images of Figure 2, 3, 4B and 5. The color images should be improved. In Figure 4B, the arrows are faint and need to be redrew.

2. In the panel D, E and F of Figure 2B, the higher expression levels of CD-Del-EGFP look like to have a less protein level of endogenous matriptase. Is it possible that CD-Del-GFP can decrease the endogenous level of matriptase?

3. In Figure 4B and 6C, there are more nuclear membrane localization for the EE→AA + L�A mutant of MTPN-EGFP than MTPN-EGFP and the other mutants. Why would this happen? Please provide a discussion to explain this phenomenon.

4. There are two grammar errors in the line 1 of the abstract and the line 6 of page 20.

6. PLOS authors have the option to publish the peer review history of their article (what does this mean?). If published, this will include your full peer review and any attached files.

Reviewer #1: No

Reviewer #2: No

---

## [Author Response · Author response to Decision Letter 0]

14 Nov 2019

Reviewer #1: 

Critiques:

I feel that the reported data do not firmly imply that the identified signal is indeed a basolateral sorting signal. Therefore, a significant amount of research has to be invested to address this point properly.

Response: 

We thank Reviewer 1 for their time and constructive comments. We agree that the data at hand do not support the assertion that the motif identified in this manuscript serves as a bone fide basolateral sorting signal. To address this concern, we have modified our language to state that the characterized seven amino acid sequence represents a motif required for matriptase translocation to the basolateral plasma membrane by entrance to the proper transport vesicles. As such, significant revisions have been made and we now present the work with new title: “The intracellular seven amino acid motif EEGEVFL is required for matriptase vesicle sorting and translocation to the basolateral plasma membrane” We believe that our data sufficiently support our revised assertion.

1). The importance of Figs 1 & 2 is to show that the ectopically expressed fusion protein localization is approximately normal. This was done by comparing its localization with respect to the endogenous protein (stained with the M24 antibody that recognizes the endogenous, but not the ectopically expressed protein) in Fig. 1, and by showing the preferential plasma membrane localization of the expressed protein with respect to ZO-1 staining, in Fig. 2. I think that the data in Fig.2 are not important for the studied question. In addition, deletion of cytoplasmic tails of transmembrane proteins typically have gross and often times artifactual effects on membrane trafficking of the mutated protein. 

Response: Although we agree with Reviewer #1 regarding the potential artefactual effects produced by the deletion of cytoplasmic tails on membrane trafficking, the deletion did abolish matriptase translocation to the plasma membrane. This result supports the important role of the cytoplasmic tail in matriptase plasma membrane targeting, regardless of the mechanism underlying the loss of membrane trafficking. Some GPI-anchored proteins contain no or very short cytoplasmic tails and can undergo co- or post-translational modifications and subsequent translocation to the plasma membrane and the apical surface. Furthermore, CD-Del-EGFP emits green fluorescent signal, suggesting that the CD-Del-EGFP construct is synthesized and does undergo somewhat normal protein folding. The loss of plasma membrane translocation may, therefore, result from preventing CD-Del-EGFP from entering either the endoplasmic reticulum (ER) or the secretory pathway. The former would mean that CD-Del-EGFP is synthesized in the cytosol rather than the ER; the latter would suggest the important role of the intracellular tail in the cell surface translocation. We, therefore, respectfully contend that while the loss of membrane targeting could result from an artefactual effect of deletion of the cytoplasmic tail, the fact that it is lost in this construct is, in fact, informative.

We have added a paragraph in the Discussion to address this concern, as follows:

“The peri-nuclear localization of CD-Del-EGFP construct and the EE→AA + L→A mutants might raise technical and mechanistic concerns regarding the cause of this localization. Although one could attribute the aberrant subcellular localization to potentially artefactual effects, the deletion and point mutations that abolish matriptase translocation to the plasma membrane or prevent the majority of the point mutated MTPN-GFP variant from translocating to the basolateral plasma membrane. This result supports the important role of the cytoplasmic tail in matriptase plasma membrane targeting and these seven amino acid residues in the basolateral targeting, regardless of the mechanism underlying the loss of plasma or basolateral membrane trafficking. Some GPI-anchored proteins, such as testisin and prostasin, contain no or a very short cytoplasmic tail and can undergo co- or post-translational modifications and subsequent translocation to the plasma membrane. Furthermore, these EGFP-containing proteins emit green fluorescent signal, and thus they do undergo protein synthesis and folding in at least a somewhat “normal” manner. The loss of plasma membrane translocation may, thus, result from preventing CD-Del-EGFP from entering either the endoplasmic reticulum (ER) or the secretory pathway. The former would mean that CD-Del-EGFP is synthesized in the cytosol rather than the ER, whereas the latter would suggest the important role of the intracellular tail in the cell surface translocation. The misrouted subcellular localization of CD-Del-EGFP has, nevertheless, its molecular and cellular basis. Similarly, the entrance into nuclei or the accumulation at the nuclear periphery for the EE→AA + L→A, the 33-39 DEL, and the EE→AA mutants, also cannot simply be interpreted as artefactual effects. The lack of the accumulation at the nuclear periphery and the nuclear localization for the L→A mutant supports the technical feasibility and soundness of the deletion and point mutations on the seven amino acid motif. The altered subcellular distribution and intracellular trafficking associated with the deletion and point mutations of the seven amino acid motif, therefore, demonstrate the important role of the motif in matriptase vesicle sorting and translocation to the basolateral plasma membrane.” 

2). Regarding data presented in Fig. 4. The resolution of this image (as well as of other images) was typically low, posing a difficulty to understand the details that the authors have pointed out in the text. For instance, I did not get their interpretation that the regions marked with arrows and appear in green mark the cell periphery. The different mutants presented show ER, or possibly some Golgi-like staining. In fact, the subtitle given to the related Figure 6 may support this idea. But, since there was not an attempt to address this point, it is hard to say where these mutants are located within the cells. I would also suggest adding the MTPN-wt distribution to the composite to facilitate the comparison. Nonetheless, this is an important point because if indeed the protein is stuck in the early secretory pathway, then the basolateral sorting signal is not at all such a signal, but an ER or Golgi localization signal. Typically, inactivation of a basolateral sorting signal causes the mislocalization of a protein to the apical surface and has no significant impact on the exit of the mutated protein from the ER or Golgi.

Response: Regarding the quality of pictures, the original Figures in Tiff file with high resolutions can be found by clicking the upper right corner of the Figure pages in the PDF file. We worry that the reviewer may have only made reference to the compressed image files present in the PDF. The deletion of all seven amino acid apparently prevented the fusion protein from translocation to the cell periphery as the EGFP was concentrated in the middle of the cells in a finely punctate pattern (Fig. 4B, 33-39Del). Similar staining patterns were also observed for the EE→AA, the L→A, and the EE→AA + L→A mutants (Fig. 4B). The lack of significant accumulation of these mutants on the cell periphery was further validated by an attempt to assess the co-localization of these mutants with the tight junction marker ZO-1 (Fig. 4B, indicated by the arrowheads), while the presence of inconspicuous yet discernible green fluorescent signal patches was occasionally seen in a small portion of the cells. Thus, while these mutants appear to be synthesized and traffic through the secretory pathway due to their perinuclear accumulation, the lack of accumulation on cell periphery suggests that these three characteristic amino acid residues play important roles in sorting and trafficking of matriptase.

3). As per the experiments described in Fig. 5, which assessed the effects of the mutated residues on the apical and basal localization of the protein in polarized MDCK cells. To verify that the mutated versions of the protein are indeed mislocated to the apical surface, the authors have to perform genuine surface stainings of the protein, using at least one of the two indicated approaches; a) performance of surface labeling of the protein using antibodies directed against the ectodomain of the protein, and under conditions whereby the cells are not permeabilized. In this strategy, providing confocal x-z images is not sufficient, and quantitative analysis of the Ap vs. Bl distribution of the protein should be provided. b) performance of cell-surface biotinylation/biochemical based assay in which cell surface proteins are biotinylated and pulled down and the protein of interest is detected by Western blotting. These experiments are fundamental for the proper addressing the effects of the mutations on the polarized distribution of matriptase.

Response: We agree with Reviewer #1 that the combination of the qualitative confocal x-z images and the quantitative analysis by surface biotinylation has been widely applied to study the apical and basolateral distribution. While the surface biotinylation approach is useful for the study of endogenous proteins, some inherent challenges present themselves when surface biotinylation is used to study the subcellular distribution of exogenously expressed proteins. When expression levels are high, the subcellular distribution of the protein of interest would be distorted to some degree. This is problematic because it is a challenge to ensure that similar exogenous protein expression levels are achieved in all cells in the experimental model. This distribution of expression levels in the cells in the system is not a problem for microscopy-based analysis, because one can simply avoid the cells that are expressing very high levels of the protein. For biochemical analyses such as cell-surface biotinylation, all cells in the culture are analyzed irrespective of their expression level, which makes interpretation of the data much more challenging as subcellular localization aberrations, present in the cells expressing supraphysiological levels of the constructs, contaminate the samples generated. We, therefore, respectfully contend that the analysis of representative x-z images from cells expressing lower levels of the constructs is the optimal approach to accurately present our results. 

4). A prominent hallmark of basolateral sorting signals is that they can act in a dominant and autonomous manner. Thus, it would be essential to show that the signal identified in this study can confer basolateral sorting of a heterologous protein that is intrinsically sorted to the apical surface, and that the mutations abrogate this process.

Response: We agree with Review #1 for the requirement for the demonstration of true autonomous basolateral sorting signals. As such, we have revised our conclusion that the seven amino acid residues identified in our study are required but not sufficient for matriptase basolateral translocation via its role in vesicle sorting. 

5). I found the paper difficult to read, containing unexplained jargon and abbreviations, s as well as long and unclear sentences (e.g., in page 11 the sentence starting with “ The lack of significant…”, or “The targeting of MTPN-EGFP to cell periphery…”).

Response: We apologize for the level of unexplained jargon, abbreviations, and run-on sentences. We have gone through the manuscript in an attempt to address these important concerns. 

 

Reviewer #2: Human matriptase is sorted to the basolateral plasma membrane via a motif with

monoleucine C-terminal to an acidic cluster

In this manuscript, Tseng CC et al addressed the question of the basolateral sorting motif for human matriptase. They identified a monoleucine C-terminal to an acidic cluster in the amino terminus of matriptase to determine matriptase sorting to the basolateral plasma membrane. The authors used a fusion protein containing a green fluorescent protein (EGFP) reporter to conjugate with the ending residue G149 in the amino terminus of matriptase. By using this fusion reporter protein, they reveal the basolateral sorting motif for matriptase. However, several issues are needed to be clarified.

1. The red and green fluorescence are not well shown in the individual images of Figure 2, 3, 4B and 5. The color images should be improved. In Figure 4B, the arrows are faint and need to be redrew.

Response: We prepared our immunofluorescent staining image of proteins in these four figures in black and white manner for the highest contrast with the exception of when pseudocolor copies of the images were merged for assessment of co-localization of the proteins to be compared. We prepare our Figures at high resolution and quality, however, the images presented in the merged PDF file are compressed and so of much lower quality. The high-quality original images can be retrieved and downloaded by clicking on the upper right corner of each figure in the PDF file. We worry that the reviewer was not aware of this arrangement and so had not seen the original high-resolution images. 

2. In the panel D, E and F of Figure 2B, the higher expression levels of CD-Del-EGFP look like to have a less protein level of endogenous matriptase. Is it possible that CD-Del-GFP can decrease the endogenous level of matriptase?

Response: It is difficult to determine whether the exogenous expression of CD-Del-GFP could affect the expression of endogenous matriptase. While stronger endogenous matriptase staining appears in the two cells right above the two CD-Del-GFP expressing cells, there seems no noticeable difference in matriptase staining in the cells at the low-left corner when compared to the two CD-Del-GFP expressing cells.

3. In Figure 4B and 6C, there are more nuclear membrane localization for the EE→AA + L�A mutant of MTPN-EGFP than MTPN-EGFP and the other mutants. Why would this happen? Please provide a discussion to explain this phenomenon.

Response: We thank the reviewer for this comment and have added a paragraph to the Discussion to address this concern, as follows:

“The peri-nuclear localization of CD-Del-EGFP construct and the EE→AA + L→A mutants might raise technical and mechanistic concerns regarding the cause of this localization. Although one could attribute the aberrant subcellular localization to potentially artefactual effects, the deletion and point mutations that abolish matriptase translocation to the plasma membrane or prevent the majority of the point mutated MTPN-GFP variant from translocating to the basolateral plasma membrane. This result supports the important role of the cytoplasmic tail in matriptase plasma membrane targeting and these seven amino acid residues in the basolateral targeting, regardless of the mechanism underlying the loss of plasma or basolateral membrane trafficking. Some GPI-anchored proteins, such as testisin and prostasin, contain no or a very short cytoplasmic tail and can undergo co- or post-translational modifications and subsequent translocation to the plasma membrane. Furthermore, these EGFP-containing proteins emit green fluorescent signal, and thus they do undergo protein synthesis and folding in at least a somewhat “normal” manner. The loss of plasma membrane translocation may, thus, result from preventing CD-Del-EGFP from entering either the endoplasmic reticulum (ER) or the secretory pathway. The former would mean that CD-Del-EGFP is synthesized in the cytosol rather than the ER, whereas the latter would suggest the important role of the intracellular tail in the cell surface translocation. The misrouted subcellular localization of CD-Del-EGFP has, nevertheless, its molecular and cellular basis. Similarly, the entrance into nuclei or the accumulation at the nuclear periphery for the EE→AA + L→A, the 33-39 DEL, and the EE→AA mutants, also cannot simply be interpreted as artefactual effects. The lack of the accumulation at the nuclear periphery and the nuclear localization for the L→A mutant supports the technical feasibility and soundness of the deletion and point mutations on the seven amino acid motif. The altered subcellular distribution and intracellular trafficking associated with the deletion and point mutations of the seven amino acid motif, therefore, demonstrate the important role of the motif in matriptase vesicle sorting and translocation to the basolateral plasma membrane.” 

4. There are two grammar errors in the line 1 of the abstract and the line 6 of page 20.

Response: We apologize for these errors, which are corrected in the revised manuscript.

---

## [Decision Letter · Decision Letter 1]

2 Jan 2020

PONE-D-19-15523R1

The intracellular seven amino acid motif EEGEVFL is required for matriptase vesicle sorting and translocation to the basolateral plasma membrane

PLOS ONE

Dear Dr. Chen-Yong Lin,

Thank you for submitting your manuscript to PLOS ONE. After careful consideration, we feel that it has merit but does not fully meet PLOS ONE’s publication criteria as it currently stands. Therefore, we invite you to submit a revised version of the manuscript that addresses the points raised during the review process.

We would appreciate receiving your revised manuscript by January 31, 2020. To enhance the reproducibility of your results, we recommend that if applicable you deposit your laboratory protocols in protocols.io, where a protocol can be assigned its own identifier (DOI) such that it can be cited independently in the future. For instructions see: http://journals.plos.org/plosone/s/submission-guidelines#loc-laboratory-protocols

We look forward to receiving your revised manuscript.

Kind regards,

Qing-Xiang Amy Sang, Ph.D.

Academic Editor

PLOS ONE

Reviewers' comments:

Reviewer's Responses to Questions

**Comments to the Author**

1. If the authors have adequately addressed your comments raised in a previous round of review and you feel that this manuscript is now acceptable for publication, you may indicate that here to bypass the “Comments to the Author” section, enter your conflict of interest statement in the “Confidential to Editor” section, and submit your "Accept" recommendation.

Reviewer #1: All comments have been addressed

Reviewer #2: (No Response)

2. Is the manuscript technically sound, and do the data support the conclusions?

Reviewer #1: Partly

Reviewer #2: Partly

3. Has the statistical analysis been performed appropriately and rigorously? 

Reviewer #1: Yes

Reviewer #2: Yes

4. Have the authors made all data underlying the findings in their manuscript fully available?

Reviewer #1: Yes

Reviewer #2: Yes

5. Is the manuscript presented in an intelligible fashion and written in standard English?

Reviewer #1: Yes

Reviewer #2: Yes

6. Review Comments to the Author

Reviewer #1: The authors should have stated the "not sufficient" statement in the manuscript and not only in the rebuttal letter. For instance, in the Abstract it should have been stated: " Our study reveals that (the word "that" was missing there) the EEGEVFL motif is necessary, but may not be sufficient, for matriptase basolateral membrane targeting....".

Reviewer #2: This is a revised manuscript. However, some points in the revised manuscript are not well addressed.

1. The real colors of the fluorescence images (EGFP, M24 and ZO-1) in Figure 2, 3, 4 and 5 were not individually shown in the revised manuscript. The image qualities in the revised manuscript are not improved.

2. In Figure 4B, some of the EGFP images were extended out of the ZO-1-stained borders. What reasons cause the phenomena? Whether those matriptase-EGFP fusion proteins will affect the formation of tight junctions?

3. Matriptase has been shown to be mainly located in the adherin junction region of polarized MDCK cells. However, MTPN-EGFP proteins looked like to be mainly co-localized with ZO-1 at the apical surface in Figure 3 and 5 of the current manuscript. It raises a question whether MTPN-EGFP can behave like a real matriptase for the analysis of the protease transcytosis. The authors should explain why MTPN-EGFP proteins did not well exhibit the real localization of matriptase in the differentiated epithelial cells?

7. PLOS authors have the option to publish the peer review history of their article (what does this mean?). If published, this will include your full peer review and any attached files.

Reviewer #1: No

Reviewer #2: No

---

## [Author Response · Author response to Decision Letter 1]

3 Jan 2020

Reviewer #1: 

Critiques:

Reviewer #1: The authors should have stated the "not sufficient" statement in the manuscript and not only in the rebuttal letter. For instance, in the Abstract it should have been stated: " Our study reveals that (the word "that" was missing there) the EEGEVFL motif is necessary, but may not be sufficient, for matriptase basolateral membrane targeting....".

Response: We agree with Reviewer #1 that although our data only support a required role of the seven amino acid motif in matriptase basolateral targeting. We have added the statement of “but may not be sufficient” in the abstract.

Reviewer #2: 

This is a revised manuscript. However, some points in the revised manuscript are not well addressed.

1. The real colors of the fluorescence images (EGFP, M24 and ZO-1) in Figure 2, 3, 4 and 5 were not individually shown in the revised manuscript. The image qualities in the revised manuscript are not improved.

Response: We do not understand what Reviewer #2 means by “real colors”. When a digital imaging system is used as herein, the images are acquired in a numerical format as intensity per pixel. Thus, the most typical representation of the data converts intensity into grayscale. “False colors” can then assigned to the black-and-white digital images by computer software. For data presentation, grayscale images give the best representation of contrast and details. When colocalizations studies are conducted for more than one species, false color representations are useful for comparisons. We respectfully contend that the individual images with false color applied will not provide better quality than the ones in black-and-white. 

2. In Figure 4B, some of the EGFP images were extended out of the ZO-1-stained borders. What reasons cause the phenomena? Whether those matriptase-EGFP fusion proteins will affect the formation of tight junctions?

Response: It is not uncommon to observe basolateral membrane ruffles in polarized cells. The ZO-1 staining defines the tight junction, however, it is not confined to the border of the cell. It has been documented that the basolateral “body” can be larger than the area defined by the tight junctions. https://www.researchgate.net/figure/Z-stack-image-analysis-of-representative-bestrophin-1-mutants-in-MDCK-II-cells-A_fig4_51606308

In addition, in Figure 5, ZO-1 was detected at the tight junction and we came to the conclusion that matriptase-EGFP does not alter tight junction formation.

3. Matriptase has been shown to be mainly located in the adherin junction region of polarized MDCK cells. However, MTPN-EGFP proteins looked like to be mainly co-localized with ZO-1 at the apical surface in Figure 3 and 5 of the current manuscript. It raises a question whether MTPN-EGFP can behave like a real matriptase for the analysis of the protease transcytosis. The authors should explain why MTPN-EGFP proteins did not well exhibit the real localization of matriptase in the differentiated epithelial cells?

Response: 

ZO-1 is a well-established tight junction marker, which was used to show the coincidence of matriptase with ZO-1 on the surface of MDCK cells in Figure 3. Our statement regarding this observation is “The peripheral staining was further confirmed by its coincidence with ZO-1, a well-established tight junction marker (Fig. 3 A, B and C, as indicated by arrowheads).” We carefully used “coincidence” rather than “co-localization” to avoid some confusions. We never stated that “MTPN-EGFP proteins looked like to be mainly co-localized with ZO-1 at the apical surface in Figure 3 and 5 of the current manuscript.” Tight junctions are not on the apical surface. It is factually incorrect for Reviewer #2 to say that ZO-1 and matriptase are colocalized on the apical surface. TMPRSS2 is on the apical surface.

The comment regarding Figure 5 further demonstrate Reviewer #2’s confusion. Should two signal occupy the same position in the XY plane, but be located at a different depth, a sample imaged at the XY orientation may show the two signals colocalizing, depending on the depth of the field (as seen in Figure 3). However, if a sample is imaged and represented in the ZY, or the ZX orientation, it would reveal that the two signals do not in fact colocalize (as seen in Figure 5). Our data showed MTPN-EGFP coincides with ZO-1 in Figure 3, because it is a micrograph imaged at the XY orientation. To further dissect the subcellular location of MTPN-EGFP, we imaged the cells at the XZ orientation, thus revealing that MTPN-EGFP is located at the basolateral surface. Our data characterizes matriptase localization at a higher resolution.

In a previous study [1], we demonstrated that endogenous matriptase was detected at the adherens junctions in 184 A1N4 human mammary epithelial cells. In another previous study [2], we showed that endogenous matriptase was detected on the basolateral plasma membrane in differentiated Caco-2 human colon adenocarcinoma cells. Our matriptase mAbs do not cross-react with canine matriptase, and to the best of our knowledge, there have been no reports that commercially available matriptase antibodies can detect canine matriptase. We, therefore, believe that Reviewer #2 must simply be mistaken in their assertion that matriptase has been shown to be mainly located in the adherins junctions of polarized MDCK cells, as we are unaware of any such study, published or presented in a scientific meeting. If by “real” matriptase, Reviewer #2 means “endogenous” matriptase in Caco-2 cells rather than in MDCK cells, the data and conclusions of our current study using MDCK cells with an engineered matriptase fusion protein are entirely consistent with our published data studying endogenous matriptase in Caco-2 cells [2]. The coincidence of endogenous matriptase and ZO-1 in the XY orientation and the differential localization of endogenous matriptase in the basolateral surface versus ZO-1 in the tight junction in ZY orientation in Caco-2 human colon adenocarcinoma cells can be found in our previous study, which is also presented here . 

Based on these data we believe that the engineered model system we have used in this study replicates the behavior of “real” matriptase with great fidelity. 

References

 1. Hung RJ, Hsu I, Dreiling JL, Lee MJ, Williams CA, Oberst MD, Dickson RB, Lin CY (2004) Assembly of adherens junctions is required for sphingosine 1-phosphate-induced matriptase accumulation and activation at mammary epithelial cell-cell contacts. Am J Physiol Cell Physiol 286: C1159-C1169.

 2. Wang JK, Lee MS, Tseng IC, Chou FP, Chen YW, Fulton A, Lee HS, Chen CJ, Johnson MD, Lin CY (2009) Polarized epithelial cells secrete matriptase as a consequence of zymogen activation and HAI-1-mediated inhibition. Am J Physiol Cell Physiol 297: C459-C470.

---

## [Decision Letter · Decision Letter 2]

27 Jan 2020

The intracellular seven amino acid motif EEGEVFL is required for matriptase vesicle sorting and translocation to the basolateral plasma membrane

PONE-D-19-15523R2

Dear Dr. Chen-Yong Lin,

We are pleased to inform you that your manuscript has been judged scientifically suitable for publication and will be formally accepted for publication once it complies with all outstanding technical requirements.

With kind regards,

Qing-Xiang Amy Sang, Ph.D.

Academic Editor

PLOS ONE

Additional Editor Comments (optional):

Reviewers' comments:

Reviewer's Responses to Questions

**Comments to the Author**

1. If the authors have adequately addressed your comments raised in a previous round of review and you feel that this manuscript is now acceptable for publication, you may indicate that here to bypass the “Comments to the Author” section, enter your conflict of interest statement in the “Confidential to Editor” section, and submit your "Accept" recommendation.

Reviewer #1: All comments have been addressed

Reviewer #2: All comments have been addressed

2. Is the manuscript technically sound, and do the data support the conclusions?

Reviewer #1: Yes

Reviewer #2: (No Response)

3. Has the statistical analysis been performed appropriately and rigorously? 

Reviewer #1: Yes

Reviewer #2: N/A

4. Have the authors made all data underlying the findings in their manuscript fully available?

Reviewer #1: Yes

Reviewer #2: Yes

5. Is the manuscript presented in an intelligible fashion and written in standard English?

Reviewer #1: Yes

Reviewer #2: Yes

6. Review Comments to the Author

Reviewer #1: (No Response)

Reviewer #2: there is no additional suggestion to the authors regarding to the manuscript: The intracellular seven amino acid motif EEGEVFL is required for matriptase vesicle sorting and translocation to the basolateral plasma membrane.

7. PLOS authors have the option to publish the peer review history of their article (what does this mean?). If published, this will include your full peer review and any attached files.

Reviewer #1: No

Reviewer #2: No

---

## [Editor Report · Acceptance letter]

29 Jan 2020

PONE-D-19-15523R2 

The intracellular seven amino acid motif EEGEVFL is required for matriptase vesicle sorting and translocation to the basolateral plasma membrane 

Dear Dr. Lin:

I am pleased to inform you that your manuscript has been deemed suitable for publication in PLOS ONE. Congratulations! Your manuscript is now with our production department. 

With kind regards,

on behalf of

Dr. Qing-Xiang Amy Sang 

Academic Editor

PLOS ONE